# UniPR1331: Small Eph/Ephrin Antagonist Beneficial in Intestinal Inflammation by Interfering with Type-B Signaling

**DOI:** 10.3390/ph14060502

**Published:** 2021-05-24

**Authors:** Carmine Giorgio, Marika Allodi, Simone Palese, Andrea Grandi, Massimiliano Tognolini, Riccardo Castelli, Alessio Lodola, Lisa Flammini, Anna Maria Cantoni, Elisabetta Barocelli, Simona Bertoni

**Affiliations:** 1Department of Food and Drug, University of Parma, Parco Area delle Scienze 27/a, 43124 Parma, Italy; carmine.giorgio@unipr.it (C.G.); marika.allodi@unipr.it (M.A.); simone.palese@studenti.unipr.it (S.P.); grandi.andrea@outlook.com (A.G.); massimiliano.tognolini@unipr.it (M.T.); riccardo.castelli@unipr.it (R.C.); alessio.lodola@unipr.it (A.L.); lisa.flammini@unipr.it (L.F.); elisabetta.barocelli@unipr.it (E.B.); 2Department of Veterinary Sciences, University of Parma, Strada del Taglio 10, 43126 Parma, Italy; annamaria.cantoni@unipr.it

**Keywords:** EphA2, ephrin-A1-Fc, TNBS-induced colitis, sulfasalazine, splenocytes

## Abstract

Eph receptors, comprising A and B classes, interact with cell-bound ephrins generating bidirectional signaling. Although mainly related to carcinogenesis and organogenesis, the role of Eph/ephrin system in inflammation is growingly acknowledged. Recently, we showed that EphA/ephrin-A proteins can modulate the acute inflammatory responses induced by mesenteric ischemia/reperfusion, while beneficial effects were granted by EphB4, acting as EphB/ephrin-B antagonist, in a murine model of Crohn’s disease (CD). Accordingly, we now aim to evaluate the effects of UniPR1331, a pan-Eph/ephrin antagonist, in TNBS-induced colitis and to ascertain whether UniPR1331 effects can be attributed to A- or B-type signaling interference. The potential anti-inflammatory action of UniPR1331 was compared to those of the recombinant proteins EphA2, a purported EphA/ephrin-A antagonist, and of ephrin-A1-Fc and EphA2-Fc, supposedly activating forward and reverse EphA/ephrin-A signaling, in murine TNBS-induced colitis and in stimulated cultured mononuclear splenocytes. UniPR1331 antagonized the inflammatory responses both in vivo, mimicking EphB4 protection, and in vitro; EphA/ephrin-A proteins were inactive or only weakly effective. Our findings represent a further proof-of-concept that blockade of EphB/ephrin-B signaling is a promising pharmacological strategy for CD management and highlight UniPR1331 as a novel drug candidate, seemingly working through the modulation of immune responses.

## 1. Introduction

Eph (erythropoietin-producing hepatocellular carcinoma) receptors belong to the largest family of tyrosine kinases receptors (RTKs): they are divided into A- (EphA1–8, 10) and B-classes (EphB1–4, 6), according to structural features and binding affinities for their ligands, the cell-surface bound ephrins (Eph receptor interacting proteins) [1]. Eph-ephrin interactions occur at cell–cell contact sites, generating a bidirectional signaling that affects both the Eph and the ephrin-bearing cells [2]: they play key roles in cell survival, proliferation, and migration and affect embryonic growth and cancer development and progression [3].

The widespread expression of both A- and B-type Eph/ephrin proteins on epithelial, endothelial and blood cells has also progressively attracted the interest of the scientific community, making them intriguing targets to tune the inflammatory responses [4,5]. Recently, our group has also focused attention on the involvement of the Eph/ephrin system in the control of acute and subacute inflammatory processes in the gut; we demonstrated that EphA/ephrin-A proteins take part in the local and remote injuries induced by mesenteric ischemia/reperfusion, an acute inflammatory condition, which is characterized by microvascular dysfunctions and by the loss of mucosal barrier integrity [6]. Conversely, type B Eph/ephrin proteins seemingly participate in the modulation of the immune responses activated in trinitrobenzene sulfonic acid (TNBS)-induced colitis [7], a conventional model of inflammatory bowel disease (IBD) with a major Th1 component.

Our interest in the Eph/ephrin system has been further boosted by the design and synthesis of UniPR1331, a small molecule developed from the chemical manipulation of litho-colic acid. This molecule acts as a potent Eph/ephrin antagonist endowed with good oral bioavailability, a pharmacokinetic property considered particularly favorable for a drug, thanks to the high compliance the oral route has in patients [8]. Accordingly, given the beneficial effects afforded by the blockade of EphB/ephrin-B signaling by monomeric EphB4 in murine TNBS-induced colitis [7], our current aim is to investigate the effects produced by UniPR1331 in the same experimental model, where the involvement of EphA/ephrin-A system is still unknown. Since the protein–protein inhibitor displays similar affinity toward type A and type B proteins, the responses evoked by UniPR1331 will be compared not only to those induced by monomeric EphB4 [7], but also to those evoked by the administration of the soluble monomeric EphA2 that, devoid of the Fc fragment, presumably interferes with EphA/ephrin-A signaling, behaving as an antagonist. The assessment of the changes induced by UniPR1331 and EphA2 on mRNA levels of EphA2 and ephrin-A1 in colitis will contribute to further enrich the scenario.

Finally, in order to better understand the role of type A proteins in intestinal inflammation, the local and systemic effects of ephrin-A1-Fc and EphA2-Fc will also be investigated both in vitro and in vivo. In fact, these recombinant chimeric proteins supposedly activate forward and reverse signaling, respectively, and in turn block the complementary signaling pathways.

On the whole, the collected results will help to clarify the targets specifically responsible for UniPR1331-mediated effects and to ascertain whether the simultaneous blockade of both Eph/ephrin signaling pathways may offer some advantages over specific interference with one of them.

## 2. Results

### 2.1. UniPR1331 Counteracted TNBS-Induced Severe Inflammatory Response

Following the development of intestinal inflammation, control (CNT) group showed a significantly higher value of the disease activity index (DAI), a parameter estimating the severity of colitis, an exacerbated mucosal injury, and exhibited a marked shortening and thickening of the colon compared to sham (S) mice not exposed to TNBS (Figure 1A–D). Instillation of the haptenating agent also provoked massive infiltration of neutrophils in the colon and in the lungs, witnessed by increased myeloperoxidase activity (MPO) in both tissues (Figure 1E–F). Treatment with UniPR1331 was able to significantly improve mice general health conditions, to dampen the mucosal damage and to attenuate the TNBS-induce neutrophil recruitment at the highest tested dose, an effect confirmed also by its ability to mitigate, although not significantly, IL-1β production in the colon (Figure 2). Similar local and systemic anti-inflammatory effects were evoked by the conventional anti-IBD drug sulfasalazine (Figure 1).

When the histological analysis of colon tissues was performed, the Eph/ephrin antagonist did not affect the extensive mucosal necrosis and the considerable granulocytes infiltration present both in the lamina propria and in the submucosa of TNBS-exposed mice (Figure 3).

### 2.2. UniPR1331 Reverted TNBS-Induced Changes in T Cells Profile

Instillation of the haptenating agent promoted the egress of CD4+ T lymphocytes from the spleen and MLNs, as demonstrated by the reduction of their percentage and of CD4+/CD8+ ratio in both lymphoid tissues of CNT mice compared to S mice. At the highest tested dose, UniPR1331 was able to fully counteract the induced changes, particularly in the spleen, while a weaker, not remarkable effect was detected in MLNs, similar to the one evoked by sulfasalazine (Figure 4).

### 2.3. Treatment with UniPR1331 Did Not Influence Colonic Eph/ephrin Gene Expression

As regards colonic Eph/ephrin gene expression, TNBS application did not change the levels of mRNA transcripts of EphA2 and ephrin-A1, since single bands of similar intensity were detected in colonic tissues of S and CNT mice; neither the treatment with UniPR1331 at 25 mg/kg nor that with the recombinant protein EphA2 affected EphA2 or ephrin-A1 gene transcription (Figure 5).

In the case of EphB4 mRNA also, whose levels, along with those of ephrin-B2, had been already presented for S, CNT and EphB4-treated mice by Grandi, Zini et al. [7], neither the instillation of the haptenating agent nor that of UniPR1331 modified their production with respect to S tissues (Figure 6A,C). Concerning ephrin-B2 gene expression, the instillation of TNBS promoted the appearance of an additional mRNA band, corresponding to a splice variant of ephrin-B2 gene, with similar intensity in vehicle-treated and in UniPR1331-treated mice (Figure 6B,D).

### 2.4. UniPR1331 Dampened TNFα Release from Splenic Mononuclear Cells

The effects of UniPR1331 were investigated also in vitro on the viability of splenic mononuclear cells and on their activation by phorbol 12-myristate 13-acetate (PMA) and ionomycin. The viability of lymphocytes was assessed by propidium iodide exclusion assay and revealed that the exposure to the Eph/ephrin antagonist up to 30 μM preserved more than 90% cell viability, a result comparable to the one obtained in basal conditions, after exposure to the vehicle (Table 1).

The stimulation by PMA and ionomycin of mononuclear cells evoked a significant release of TNF-α in the cell culture media; following the exposure to increasing concentrations of UniPR1331, the production of the pro-inflammatory cytokine by immune cells was progressively attenuated, reaching the lowest levels when UniPR1331 was applied at 30 μM (Figure 7A).

### 2.5. EphA2 Did Not Affect the In Vitro and In Vivo Inflammatory Responses

Given the ability of UniPR1331 to block both EphA and EphB receptors, the effects of monomeric EphA2, supposedly interfering with EphA/ephrin-A signaling, as EphB4 does with EphB/ephrin-B pathways, were tested in the same experimental models in vitro and in vivo. Indeed, when applied at 1μg/mL, a concentration devoid of effects on lymphocytes viability (Table 1) and 10 times higher than that of EphB4 able to significantly increase the cytokine production [7], EphA2 was not able to modify the TNFα release by splenic mononuclear cells (Figure 7B). In vivo, the administration of EphA2 was ineffective in counteracting the inflammatory responses induced by TNBS instillation at the equimolar dose of EphB4 (Figure 8 and Table 2), nor did it modify EphA2 and ephrin-A1 genes transcription in the inflamed colon (Figure 5).

### 2.6. Ephrin-A1-Fc Evoked Weak Protective Effects against TNBS-Induced Inflammation

In order to proceed with the investigation on the role of the EphA/ephrin-A system in intestinal inflammation, we tested the effects of the recombinant proteins in TNBS-induced colitis. The activation of reverse signaling by EphA2-Fc was devoid of remarkable anti-inflammatory effects (Figure 8), while the treatment with ephrin-A1-Fc significantly improved mice general conditions (Figure 8A). Accordingly, we decided also to test a higher dose of the recombinant protein: the results are presented in Appendix A and confirmed the beneficial action of both doses of ephrin-A1-Fc on mice health status and the ability of the higher dose to attenuate, although not significantly, the colonic mucosal lesions and the recruitment of neutrophils in the colon and in the lungs. The altered profile of splenic and MLN T cells after TNBS instillation was not modified by the pharmacological modulation of the EphA/ephrin-A system (Table 2). To complete the picture, the effects of ephrin-A1-Fc and EphA2-Fc were tested also on the activation of splenic mononuclear cells and revealed that both were able to dampen the release of TNFα by about 30% at concentrations not affecting the cells’ viability (Figure 7C and Table 1).

## 3. Discussion

Following a line of research begun a few years ago, the main aim of the present study was to investigate the effects exerted by the pharmacological modulation of the Eph/ephrin system in subacute intestinal inflammation. By applying two different models of experimental colitis, we demonstrated that the blockade of the EphB/ephrin-B system could provide beneficial effects against the gut inflammation mainly elicited by Th1-mediated immune responses (TNBS-induced colitis), while being essentially devoid of effects against the colitis usually ascribed to innate mechanisms (DSS-induced colitis) [7]. These findings and the availability of UniPR1331, a pan-Eph/ephrin antagonist, endowed with high potency and advantageous pharmacokinetic features [8], tailored the main premises of the present work. In particular, the ability of UniPR1331 to efficaciously interfere with Eph/ephrin signaling pathways in vitro [8] prompted us to assess its in vitro and in vivo effects and to compare them with the effects induced by the administration of monomeric EphA2: like EphB4, monomeric EphA2 is devoid of the Fc fragment and presumably interferes with EphA/ephrin-A signaling, behaving as an antagonist.

The collected results showed that UniPR1331 displayed a remarkable anti-inflammatory action in vivo, significantly improving mice health status, reducing colonic macroscopic damage and structural alterations and counteracting local and lung neutrophil infiltration. These effects closely resemble those of sulfasalazine, conventional anti-IBD agent endowed with pronounced anti-flogistic local actions; moreover, given the oral bioavailability of UniPR1331, we can speculate that its beneficial effects probably ensue also from of its systemic absorption. Finally, the ability of UniPR1331 to counteract also the changes induced by the haptenating agent on splenic T cells profile, at variance with sulfasalazine, which is ineffective in this respect, suggests that the protection afforded by the Eph/ephrin antagonist also unfolds through the modulation of adaptive immune responses. Particularly relevant is the fact that the in vivo actions of UniPR1331 are perfectly mimicked by EphB4 and strikingly oppose the lack of activity demonstrated by equimolar EphA2.

We could also notice that the flogistic process triggered by TNBS apparently did not affect the transcription of EphA2 and ephrin-A1 genes in mice colon: this result differs from our previous observations of an alternative ephrin-B2 gene variant promoted by the intestinal inflammation in the same experimental model [7], observations in their turn consistent with the up-regulation of ephrin-B2 mRNA in the mucosal lesions of CD patients [9]. In the case of the shorter variant of ephrin-B2 gene, whose appearance is elicited by the inflammatory environment and not modified by the treatment with the small molecule, the effect of UniPR1331 also overlapped with that of EphB4. As already discussed for the recombinant protein [7], it is probable that the anti-inflammatory actions exhibited by EphB antagonists rely on down-stream factors not directly involved in the transcriptional regulation of ephrin-B2 gene. As regards A-type signaling pathway, the collected results suggest, otherwise, that it has a minor involvement in colitis pathogenesis, although the expression of EphA/ephrin-A proteins was documented in basal conditions in the human colonic crypts and tops [10].

These findings give us enough confidence to conclude that the in vivo activity of UniPR1331 in TNBS-induced colitis could be attributed to the blockade of endogenous EphB/ephrin-B signaling, while the interference with A-type pathway as a possible mechanism of action might be presumably ruled out.

Given the key role of the immune cells in the pathogenesis of IBD and of TNBS-induced colitis and the ability of UniPR1331 to restore the profile of splenic T cell subpopulations, UniPR1331 effects were investigated also on TNFα release by stimulated mononuclear splenocytes and, surprisingly, the behaviour of the small molecule stood out from that of both monomeric proteins. In fact, UniPR1331 was able to depress the production of the inflammatory cytokine in a concentration-dependent manner and, in so doing, differed both from EphA2, devoid of any effect, and from EphB4, able to potentiate immune cells activation and TNFα production [7]. If, on a mechanistic point of view, the in vitro data further supported the anti-inflammatory action exerted in vivo by UniPR1331, its molecular target remained elusive. Indeed, on the basis of the gathered data, the effects on immune cells seem to be not directly mediated by the interference with the Eph/ephrin signaling system, raising the possibility that an additional binding site is involved. The selectivity of UniPR1331 had already been deeply investigated and the interaction with various targets of steroidal derivatives, like TGR5 and PXR [8], or with targets promoting cell proliferation or angiogenesis [11] had been excluded up to the concentration of 10 μM. However, a quite intriguing aspect is the documented ability of lithocholic acid, prototypical compound from which UniPR1331 stemmed, to interfere with adaptive immune responses: in particular, lithocholic acid was reported to hamper Th1 differentiation and the release of pro-inflammatory cytokines by Jurkat T cells and human/mouse CD4^+^ T cells [12,13]. Although unlikely, given the differences existing in the structure of the rigid core between the bile acid and the cholenic derivative UniPR1331, it is appealing to hypothesize that the anti-inflammatory action of UniPR1331 may ensue both from EphB/ephrin-B blockade and from the contribution of a component related to lithocholic acid.

Further exploring the participation of EphA/ephrin-A proteins in TNBS-induced colitis, despite the fact that the blockade of this signaling pathway appeared irrelevant in colitis responses, the stimulation of both forward and reverse A-signaling was able to mitigate the production of TNFα by mononuclear splenocytes. This effect disclosed a possible addition of these pathways to the modulation of inflammation. Indeed, we could observe that the in vivo exogenous activation of EphA2 forward signaling by ephrin-A1-Fc was effective in improving mice health status and in slightly attenuating the recruitment of neutrophils in local and remote tissues in colitis, effects recalling the protective action displayed by ephrin-A1-Fc in mesenteric I/R [6]. On its turn, ephrin-A1 reverse signaling by EphA2-Fc seemed to mitigate the extracellular matrix deposition elicited in the colon by the subacute inflammatory responses. Provided that the effects on TNFα release by isolated mononuclear splenocytes are not necessarily suggestive of an in vivo impact, these findings allow us to speculate that the moderate protection given by exogenous unidirectional activation of EphA/ephrin-A signaling may depend also on limiting the responses of immune cells, subsequently attenuating the release of inflammatory cytokines. This action could contribute to hamper the enrolment of neutrophils and also to mitigate the triggered pro-fibrotic responses in TNBS-induced colitis, although, up to now, the available literature on the subject is quite scanty.

In conclusion, UniPR1331 emerged as a very promising small molecule Eph/ephrin antagonist, able to exert relevant benefits after oral dosing in a model of intestinal inflammation mainly driven by Th1 responses: the ability to potently interfere with EphB/ephrin-B signaling appears as a key factor and adds to the already described remarkable anticancer properties of the molecule [11]. Future studies will be pivotal to achieve a more in-depth picture of the beneficial actions of UniPR1331 and of its potential therapeutic applications in diseases where inflammation and tumorigenesis are involved.

## 4. Materials and Methods

### 4.1. Animals

C57BL/6 mice (8–12 weeks old) (Charles River Laboratories, Calco, Italy), weighing 20–24 g, were housed five per cage in identical conditions for at least seven days before experiments started. They were maintained under standard conditions at our animal facility (12:12 h light–dark cycle, 22–24 °C, food and water available ad libitum). All the experimental procedures and the suppression by CO_2_ inhalation were performed between 9 a.m. and 12 a.m. All appropriate measures were taken to minimize pain or discomfort of animals. Animal experiments were performed according to the guidelines for the use and care of laboratory animals and they were authorized by the local Animal Care Committee “Organismo Preposto al Benessere degli Animali” and by Italian Ministry of Health “Ministero della Salute” (Authorization n. 826/2018-PR).

### 4.2. Induction of Colitis

Intrarectal (i.r.) instillation of 50 μL of a 10% (*w*/*v*) TNBS solution in 50% ethanol was performed through insertion of a 10-cm long PE-50 tubing attached to a tuberculin syringe 3 cm into the anus of mice. Inoculated mice were kept in a vertical head-down position for 3 min to avoid leakage of the haptenating agent.

### 4.3. Experimental Design

Pharmacological treatments started 8 h after the induction of colitis and were applied once or twice daily (UniPR1331), 8 h apart, during the following two days, by subcutaneous (s.c.) (recombinant proteins) or per os (p.o.) (UniPR1331 and sulfasalazine) administration. Animals were suppressed by CO_2_ inhalation three days after TNBS or saline instillation.

Mice were assigned through block randomization to the sham group (S), comprising mice intra-rectally inoculated with 50 μL 0.9% NaCl (saline solution) and administered 10 mL/kg saline s.c. (*n* = 20), or to the following experimental groups of colitic mice: saline (CNT, 10 mL/kg, s.c., *n* = 30), UniPR1331 10 mg/kg b.i.d. (Uni10) (*n* = 10) and 25 mg/kg b.i.d. (Uni25) (*n* = 20), ephrin-A1-Fc 16 μg/kg (eph16) (*n* = 7) and 50 μg/kg (eph50) (*n* = 7), EphA2 20 μg/kg (Eph) (*n* = 12), EphA2-Fc 30 μg/kg (Eph-Fc) (*n* = 6), sulfasalazine 50 mg/kg (Sfz) (*n* = 7).

The applied doses of EphA/ephrinA recombinant proteins were equimolar to EphB/ephrinB proteins administered in a previous investigation [7], the dosage of UniPR1331 was chosen on the basis of Festuccia et al., 2018 [11], and the dose of sulfasalazine was chosen according to Grandi et al., 2017 [14]. The study was performed using experimental blocks composed by 10 or 12 mice that were randomly assigned to four or five groups of treatment (S and CNT were present in each experimental block), each one encompassing two animals.

Due to possible seasonal variability, S and CNT mice were repeated periodically all through the study in order to verify the attainment of a constant degree of colitis severity with respect to physiological conditions, thus explaining the bigger size of S and CNT experimental groups with respect to the other groups. Accordingly, each group of animals was randomly subdivided in two subgroups: colons excised from each subset were reserved either for histological analysis or for myeloperoxidase (MPO) activity determination and for cytokine assays.

### 4.4. Evaluation of Inflammatory Responses

Disease Activity Index (DAI) was measured daily throughout the experimentation. Immediately after suppression the macroscopic damage of colonic mucosa was assessed as macroscopic score (MS). The wet weight and the length of each colon were measured and the weight/length ratio was considered as disease-related intestinal wall thickening. Colon, lungs, spleen and mesenteric lymph nodes were collected for subsequent microscopic, biochemical or flow cytometry analyses.

#### 4.4.1. Disease Activity Index (DAI)

The severity of experimental colitis was estimated as DAI, a score assigned daily on the basis of body weight loss, stool consistency and rectal bleeding, by unaware investigators in accordance to Cooper’s modified method [15]. The scores (max = 9) were quantified as follows: stool consistency: 0 (normal), 1 (soft), 2 (liquid); body weight loss: 0 (<5%), 1 (5–10%), 2 (10–15%), 3 (15–20%), 4 (20–25%), 5 (>25%); rectal bleeding: 0 (absent), 1 (light bleeding), 2 (heavy bleeding).

#### 4.4.2. Macroscopic Damage of Colonic Mucosa (MS)

After suppression, the colon was explanted, opened longitudinally, gently flushed with saline solution and MS was assessed through examination of the mucosa by an investigator unaware of the treatments applied. MS was determined according to previously published criteria [16] as the sum of scores (max = 14) attributed as follows: presence of strictures and hypertrophic zones (0, absent; 1, 1 stricture; 2, 2 strictures; 3, more than 2 strictures); mucus (0, absent; 1, present); adhesion areas between the colon and other intra-abdominal organs (0, absent; 1, 1 adhesion area; 2, 2 adhesion areas; 3, more than 2 adhesion areas); intraluminal hemorrhage (0, absent; 1, present); erythema (0, absent; 1, presence of a crimsoned area < 1 cm^2^; 2, presence of a crimsoned area > 1 cm^2^); ulcerations and necrotic areas (0, absent; 1, presence of a necrotic area < 0.5 cm^2^; 2, presence of a necrotic area > 0.5 cm^2^ and <1 cm^2^; 3, presence of a necrotic area > 1 cm^2^ and <1.5 cm^2^; 4, presence of a necrotic area >1.5 cm^2^).

#### 4.4.3. Colonic Length and Thickness

The length of the colon and its weight were measured to assess deposition of fibrotic material and muscular contraction elicited by colitis induction; weight/length ratio was calculated to assess colon thickness, according to previously published criteria [14].

#### 4.4.4. Colonic and Pulmonary Myeloperoxidase (MPO) Activity Assay

MPO activity, a marker of granulocytic infiltration within a tissue, was determined according to Ivey’s modified method [17]. After being weighed, each colonic or lung sample was homogenized in ice-cold 0.02 M sodium phosphate buffer (pH 4.7), containing 0.015 M Na_2_EDTA and 1 µg mL^−1^ aprotinin, and centrifuged for 20 min at 12,500 RCF at 4 °C. Pellets were re-homogenized in four volumes of ice-cold 0.2 M sodium phosphate buffer (pH 5.4) containing 0.5% hexadecylthrimethyl-ammoniumbromide (HTAB) and 1 µg mL^−1^ aprotinin. Samples were then subjected to three cycles of freezing and thawing and centrifuged for 30 min at 15,500 RCF at 4 °C. Then, 50 µL of the supernatant was allowed to react with 950 µL of 0.2 M sodium phosphate buffer, containing 1.6 mM tetramethylbenzidine, 0.3 mM H_2_O_2_, 12% dimethyl formamide, 40% Dulbecco’s phosphate buffered saline (PBS). Each assay was performed in duplicate and the rate of change in absorbance was measured spectrophotometrically at 690 nm (TECAN Sunrise™ powered by Magellan™ data analysis software, Mannedorf, Switzerland). 1 unit of MPO was defined as the quantity of enzyme degrading 1 μmol of peroxide per minute at 25 °C. Data were normalized with edema values ((wet weight-dry weight)/dry weight) [18] and expressed as U/g of dry weight tissue.

#### 4.4.5. Colonic IL-1β Levels

In colonic samples collected from S mice (*n* = 6) and TNBS-treated animals administered with saline (*n* = 10) or UniPR1331 25 mg/kg (*n* = 7), colonic IL-1β levels were determined using a commercially available ELISA kit (IL-1β Mouse SimpleStep ELISA™ kit, Abcam Biochemicals™, Cambridge, UK). Samples were homogenized for 1 min in 700 μL of extraction buffer in accordance to the manufacturer’s protocols. Samples were then centrifuged for 30 min at 14,000 RCF and the supernatant was collected. Total protein concentration was quantified using Pierce BCA protein assay kit (ThermoFisher Scientific Inc., Waltham, MA, USA). Colonic concentrations of IL-1β were determined in duplicate in 100 µL samples: the absorbance was measured spectrophotometrically at 450 nm (TECAN Sunrise™ powered by Magellan™ data analysis software, Mannedorf, Switzerland). The assays sensitivity for IL-1β was 5 pg/mL. Results were expressed as ng/g protein.

#### 4.4.6. Colonic Histology

Colonic samples were harvested from S mice (*n* = 3) and TNBS-treated animals administered with saline (*n* = 3) or UniPR1331 25 mg/kg (*n* = 4), immersion-fixed in 10% neutral buffered formalin overnight, dehydrated and embedded in paraffin. For each sample, at least five transverse 5-µm sections were cut in the distal colon, stained with hematoxylin-eosin and blindly examined in a light microscope (Nikon Eclipse E800). The histological damage was quantified using Bischoff’s modified method [19]: the grade of mucosal destruction (0, normal; 1, mild; 2, moderate; 3, severe) and the degree of leukocytes infiltration in the lamina propria and submucosa (0, absent; 1, mild; 2, pronounced) were scored (maximum total score: 7). The average value of histological score was determined for each colon, pooled with those determined for colons of the other animals in the same experimental group and the median value was calculated.

#### 4.4.7. Reverse Transcription Polymerase Chain Reaction (RT-PCR)

Total RNA was isolated from colonic samples of S and TNBS-induced colitic mice, either treated with saline, UniPR1331 25 mg/kg or EphA2 20 μg/kg through a Qiagen RNeasy Protect Mini Kit (Qiagen, Hilden, Germany) and quantified using Nanodrop ND-1000 (Thermo Fisher Scientific Inc, Waltham, MA, USA). Next, 1 μg of RNA was reverse transcribed into cDNA and amplified using OneStep RT-PCR kit (Qiagen, Hilden, Germany), according to the manufacturer’s protocol. The following primer pairs, purchased by Life Technologies Italia (Monza, MB, Italy), were used:

-EphA2, 5′-GAGTGTCCAGAGCATACCCT-3′ (forward),

5′-GCGGTAGGTGACTTCGTACT-3′ (reverse);

-ephrin-A1, 5′-CATCATCTGCCCACATTACG-3′ (forward),

5′-AGCAGTGGTAGGAGCAATAC-3′ (reverse);

-EphB4, 5′-AGCCCCAAATAGGAGACGAG-3′ (forward),

5′-GGATAGCCCATGACAGGATC-3′ (reverse);

-ephrin-B2, 5′-ACCCACAGATAGGAGACAAA-3′ (forward),

5′-GGTTGATCCAGCAGAACTTG-3′ (reverse);

-GAPDH, 5′-GACTCCACTCACGGCAAATT-3′ (forward),

5′-TCCTCAGTGTAGCCCAAGAT-3′ (reverse).

PCR was conducted for 36 cycles to amplify Eph/ephrin cDNA sequences according to Ogawa et al., 2006 [20] and Mukai et al., 2017 [21]. The following conditions were used for amplification: EphA2: denaturation for 45 s at 94 °C, annealing for 45 s at 62.5 °C, extension for 60 s at 72 °C; ephrin-A1: denaturation for 45 s at 94 °C, annealing for 45 s at 60 °C, extension for 60 s at 72 °C; EphB4: denaturation for 10 s at 94 °C, annealing for 45 s at 53 °C and extension for 3 min at 68 °C; ephrinB2: denaturation for 45 s at 94 °C, annealing for 45 s at 53 °C and extension for 90 s at 72 °C. PCR products were separated on 1% agarose gels, visualized with a ChemiDoc™ Imaging System (Bio-Rad, Berkeley, CA, USA) after staining with RedSafe™ (iNtRON Biotechnology, Seongnam, Korea) and analysed by means of Image Lab™ software (version 6.0, Bio-Rad Laboratories, Inc., CA, USA). Expression levels of Eph/ephrin mRNAs were determined from 5–7 independent samples after normalization by reference to the signal intensity of the band of GAPDH mRNA (PCR for 30 cycles, denaturation for 45 s at 94 °C, annealing for 45 s at 93 °C, extension for 90 s at 72 °C) obtained in the same specimen.

#### 4.4.8. Flow Cytometry Assays

Isolation of Splenocytes

After suppression, the spleen from mice of the various experimental groups was removed, mechanically dispersed through a 100 μm cell-strainer, and washed with PBS containing 0.6 mM EDTA (PBS-EDTA). The cellular suspension was then centrifuged at 1500 RCF for 10 min at 4 °C, the pellet re-suspended in PBS-EDTA, incubated with 2 mL of NH_4_Cl lysis buffer (0.15 M NH_4_Cl, 1 mM KHCO_3_, 0.1 mM EDTA in distilled water) for 5 min, in the dark, to provoke erythrocytes lysis and centrifuged at 1500 RCF for 10 min at 4 °C. Then, pellets were washed with PBS-EDTA and re-suspended in 5 mL cell staining buffer (PBS containing 0.5% fetal calf serum (FCS) and 0.1% sodium azide). Finally, the cellular suspension was stained with fluorescent antibodies [22].

Isolation of Mesenteric Lymph Nodes (MLN)

After suppression, harvesting of the whole MLN chain located in the mesentery of proximal colon was performed. The explanted tissue was rinsed with PBS, vascular and adipose tissues were removed to isolate MLN, mechanically dispersed through a 100 μm cell-strainer and washed with Hank’s Balanced Salt Solution (HBSS) containing 5% FCS. The obtained suspension was centrifuged at 1500 RCF for 10 min at 4 °C, and the pellet was washed with HBSS containing 5% FCS and re-suspended in 3 mL cell staining buffer. Finally, the cellular suspension was stained with fluorescent antibodies.

Immunofluorescent Staining

Prior to staining with antibodies, 200 µL of cellular suspension was incubated with IgG1-Fc (1 µg/10^6^ cells) for 10 min in the dark at 4 °C to block non-specific binding sites for antibodies. The following antibodies were used for fluorescent staining: Phycoerythrin-Cyanine 5 (PE-Cy5) conjugated anti-mouse CD3ε (0.25 µg/10^6^ cells, catalog number 15-0031, lot number B226301), Fluorescein Isothiocyanate (FITC) anti-mouse CD4 (0.25 µg/10^6^ cells, catalog number 100406, lot number B210488), PE anti-mouse CD8a (0.25 µg/10^6^ cells, catalog number 100708, lot number B190687). Cells were incubated with antibodies for 1 h in the dark at 4 °C, washed with PBS to remove excessive antibody and suspended in cell staining buffer to perform flow cytometry analysis. The viability of the cellular suspension was determined through propidium iodide (PI) staining, a membrane impermeable fluorescent dye, excluded by viable cells, that binds to DNA emitting red fluorescence, thus resulting as a suitable marker for dead cells. Cells were incubated with PI 10 µg/mL for 1 min in the dark, at room temperature, and immediately subjected to flow cytometry analysis. Only PI^−ve^ cells were included in the analysis.

Samples were analysed using InCyte™ software (Merck Millipore, Darmstadt, Germany). Cell populations were defined as follows: lymphocytes gated in the Forward Scatter (FSC)-Side Scatter (SSC) plot (FSC low: SSC low); T lymphocytes (CD3^+^ lymphocytes); CD4^+^ helper T lymphocytes (CD3^+^CD4^+^CD8^−^ lymphocytes); CD8^+^cytotoxic T lymphocytes (CD3^+^CD8^+^CD4^−^ lymphocytes). Percentages of CD4^+^ T helper lymphocytes and CD8^+^ T cytotoxic lymphocytes with respect to CD3^+^ lymphocytes and the ratio between CD4^+^ T helper and CD8^+^ T cytotoxic lymphocytes were calculated.

### 4.5. In Vitro Assays

#### 4.5.1. Mononuclear Cells Culture

Spleens were removed from naïve mice and splenocytes were isolated as previously described. Splenocyte suspensions were subjected to 40–80% Percoll^®^ (GE-Healthcare, Chicago, IL, USA) density gradient centrifugation. Cells were spun at 1000 RCF for 20 min at 20 °C and mononuclear cells, comprising lymphocytes and monocytes, were collected from the 40–80% interface, washed with RPMI-1640 (EuroClone, Pero, MI, Italy) and re-suspended in medium containing heat-inactivated 10% fetal bovine serum (FBS) (Gibco, Carlsbad, CA USA) [7]. Cells were then plated at 10^6^ cells/mL in 96-well plates and cultured for 24 hrs at 37 °C in a humidified atmosphere with 5% CO_2_. Ionomycin (500 ng/mL) and phorbol 12-myristate 13-acetate (PMA) (50 ng/mL) were added to the culture media in the final 4 hrs of incubation to stimulate cells in the absence or presence of UniPR1331 3–30 μM, IgG1-Fc 0.3 μg/mL, ephrin-A1-Fc 0.5 μg/mL, EphA2-Fc 1 μg/mL, and EphA2 1 μg/mL [23].

At the end of the incubation period, the culture media were frozen at −80 °C until cytokine measurement and assessment of cell viability by propidium iodide exclusion.

#### 4.5.2. Determination of TNF-α Levels in Cultured Mononuclear Cells

TNF-α levels were determined in duplicate in 100 µL samples, using a commercially available ELISA kit (Mouse TNF alpha ELISA Ready-SET-Go!, eBioscience™, San Diego, CA, USA). The absorbance was measured spectrophotometrically at 450 nm (TECAN Sunrise™ powered by Magellan™ data analysis software, Mannedorf, Switzerland). The assays sensitivity was 8 pg/mL.

#### 4.5.3. Lymphocytes Viability by Propidium Iodide Exclusion

Mononuclear cells were incubated with PI 10 µg/mL for 1 min in the dark, at room temperature, and immediately subjected to flow cytometry analysis: viable cells were identified as cells negative for the red fluorescence. The percentage of viability among lymphocytes, identified through the gating in the Forward Scatter (FSC)-Side Scatter (SSC) plot (FSC low: SSC low), was determined.

### 4.6. Statistics

All data were presented as means ± SEM. Comparison among experimental groups were made using analysis of variance (one-way or two-way ANOVA) followed by Bonferroni’s post-test, when *p* < 0.05, chosen as level of statistical significance, was achieved. Non-parametric Kruskal-Wallis analysis, followed by Dunn’s post-test, was applied for statistical comparison of MS or in case of non-normal distribution of the data. All analyses were performed using Prism 4 software (GraphPad Software Inc. San Diego, CA, USA).

### 4.7. Drugs, Antibodies and Reagents

Recombinant mouse EphA2-Fc chimera and ephrinA1-Fc chimera were purchased from R&D systems™ (Minneapolis, MN, USA), mouse EphA2 (His Tag) from Sino Biological Inc.™ (Beijing, China). FITC anti-mouse CD4, PE anti-mouse CD8 and propidium iodide were purchased from BioLegend™ (San Diego, CA, USA), PE-Cy5 anti-mouse CD3 from Affymetrix eBioscience™ (San Diego, CA, USA) and IgG1-Fc from Millipore™ (Merck, Darmstadt, Germany). TNBS, ethanol, HTAB, hydrogen peroxide, tetramethylbenzidine, dimethyl formamide and all the other reagents were purchased from Sigma Aldrich™ (St. Louis, MO, USA). On the day of the experiment, recombinant proteins were dissolved in saline solution and UniPR1331 and sulfasalazine were dispersed in carboxymethylcellulose 0.5%. UniPR1331 was dissolved in DMSO 0.3% when tested in in vitro assays.

## Figures and Tables

**Figure 1 pharmaceuticals-14-00502-f001:**
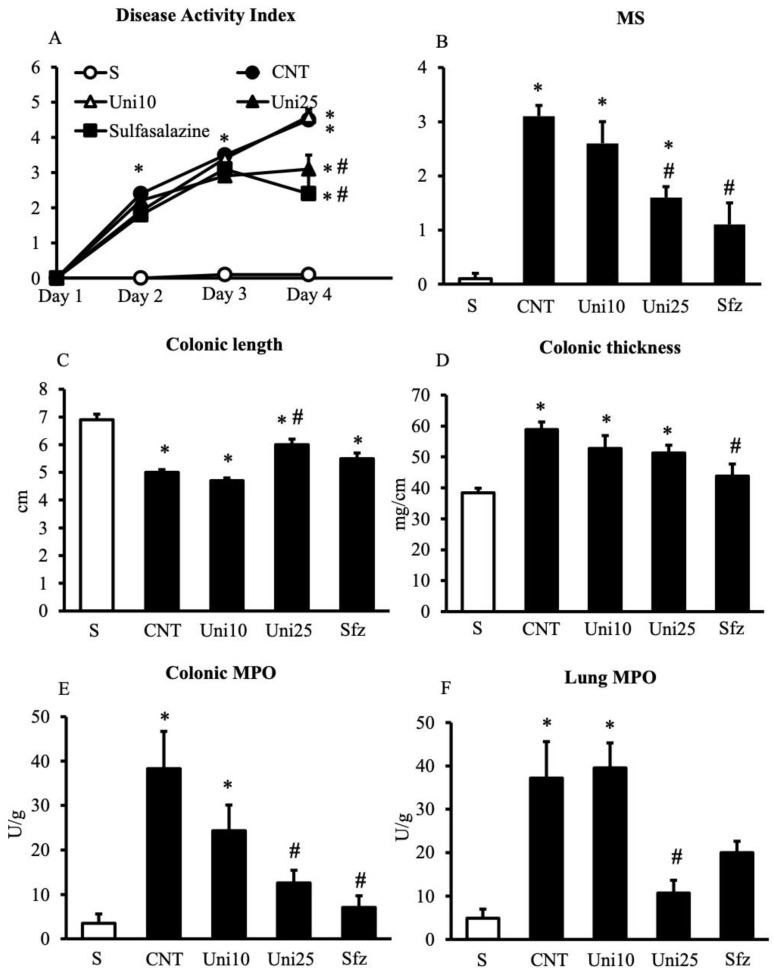
Effects of UniPR1331 on TNBS-induced inflammatory responses. Disease Activity Index (**A**), macroscopic score (MS) (**B**), colonic length (**C**), colonic thickness (**D**), colonic myeloperoxidase (MPO) (**E**) and lung MPO (**F**) activity assessed in vehicle-treated normal mice (S) and in TNBS-treated mice administered with vehicle (CNT), UniPR1331 10 mg/kg (Uni10), UniPR1331 25 mg/kg (Uni25), and sulfasalazine 50 mg/kg (Sfz) (*n* = 5–15 independent values per group). * *p* < 0.05 vs. S mice; ^#^ *p* < 0.05 vs. CNT mice; one-way or two-way (DAI) ANOVA followed by Bonferroni’s post-test; Kruskal-Wallis followed by Dunn’s post-test (MS).

**Figure 2 pharmaceuticals-14-00502-f002:**
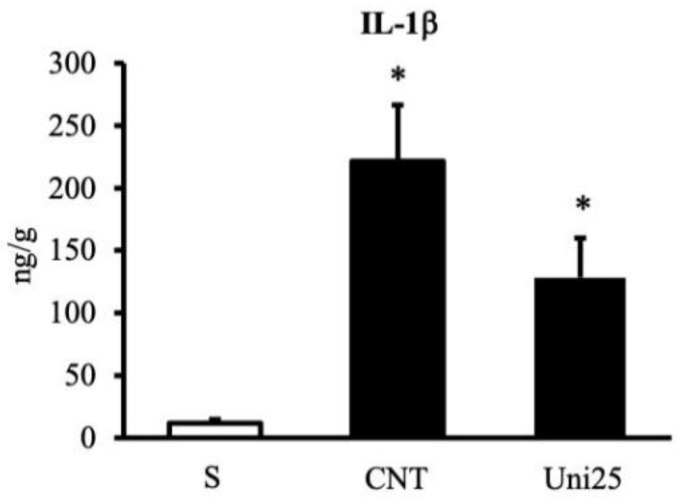
UniPR1331 attenuated TNBS-induced increase in IL-1β. IL-1β levels in colonic tissues excised from vehicle-treated normal mice (S) and TNBS-treated mice administered with vehicle (CNT) or UniPR1331 25 mg/kg (Uni25) (*n* = 6–10 independent values per group). * *p* < 0.05 vs. S mice, Kruskal-Wallis test followed by Dunn’s post-test.

**Figure 3 pharmaceuticals-14-00502-f003:**
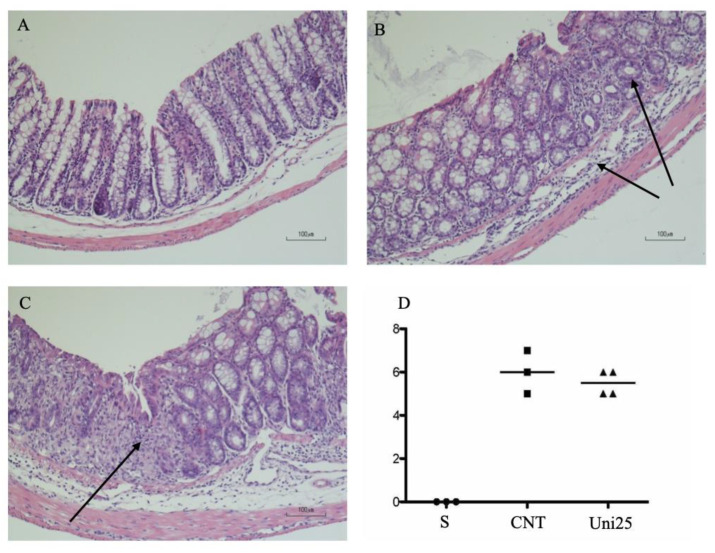
Representative hematoxylin-eosin stained sections of colonic specimens harvested from vehicle-treated S mice (**A**) and from TNBS-treated mice administered with vehicle (**B**) or UniPR1331 25 mg/kg (**C**). TNBS colonic instillation caused neutrophils infiltration and submucosal oedema (indicated by arrows) in vehicle-treated animals not attenuated in UniPR1331-treated mice (**C**). Panel (**D**) represents histological damage scoring of colonic sections obtained from vehicle-treated S mice (●), CNT mice (■) or UniPR1331-treated colitic mice (▲) (horizontal bar at the median value).

**Figure 4 pharmaceuticals-14-00502-f004:**
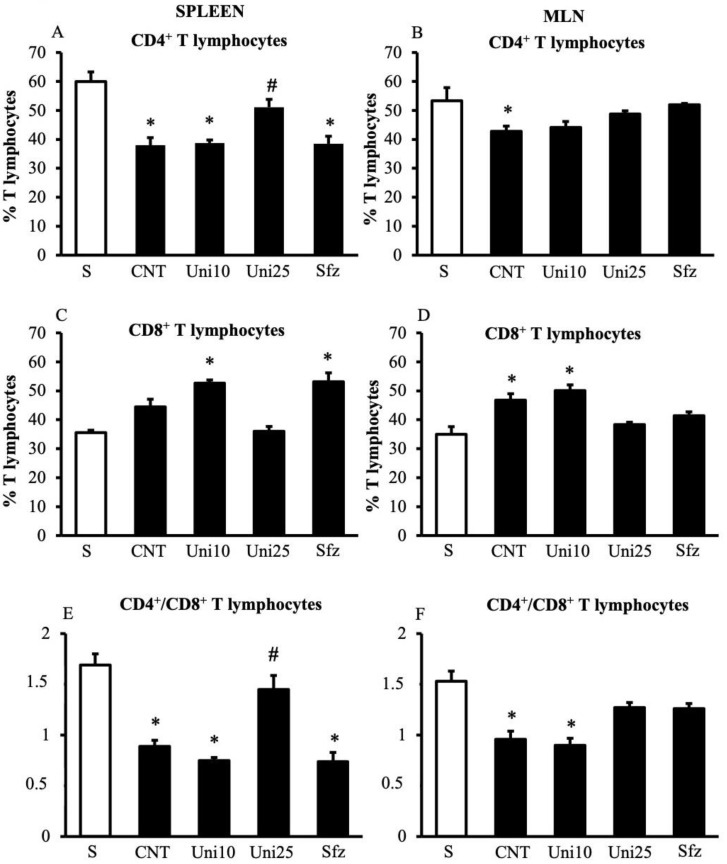
UniPR1331 mitigated TNBS-induced changes in spleen T cells profile. Percentages of CD4+ (**A**,**B**), CD8+ (**C**,**D**), and the ratio between CD4+ and CD8+ T lymphocytes (**E**,**F**) were assessed in the spleen (**A**,**C**,**E**) and MLN (**B**,**D**,**F**) excised from vehicle-treated normal mice (S) and TNBS-treated mice administered with vehicle (CNT), UniPR1331 10 mg/kg (Uni10) or 25 mg/kg (Uni25), or sulfasalazine 50 mg/kg (Sfz) (*n* = 5–15 independent values per group). * *p* < 0.05 vs. S mice; ^#^ *p* < 0.05 vs. CNT mice, one-way ANOVA followed by Bonferroni’s post-test.

**Figure 5 pharmaceuticals-14-00502-f005:**
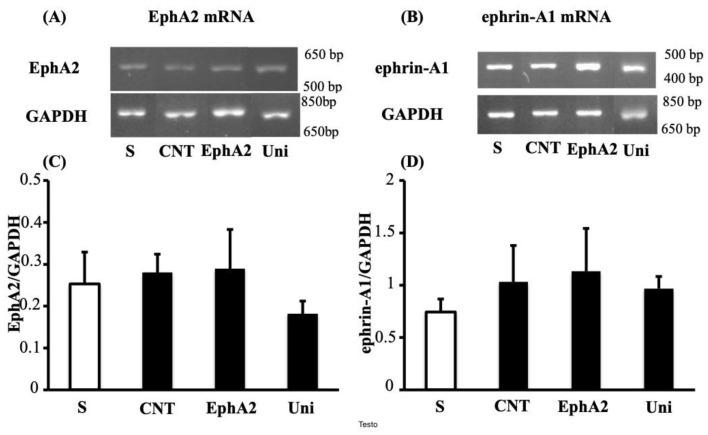
UniPR1331 and EphA2 did not modify colonic levels of EphA2 and ephrin-A1 mRNA. Representative agarose gels showing mRNA levels of EphA2 (**A**), ephrin-A1 (**B**) and GAPDH from the colon of vehicle-treated normal mice (S; *n* = 5) and of TNBS-treated mice administered with vehicle (CNT; *n* = 5), EphA2 20 mg/kg (EphA2; *n* = 5) or UniPR1331 25 mg/kg (Uni; *n* = 5). Histograms represent the quantification of EphA2 (**C**) and ephrin-A1 (**D**) mRNA levels normalized to GAPDH amplification products. Data are shown as mean ± SEM.

**Figure 6 pharmaceuticals-14-00502-f006:**
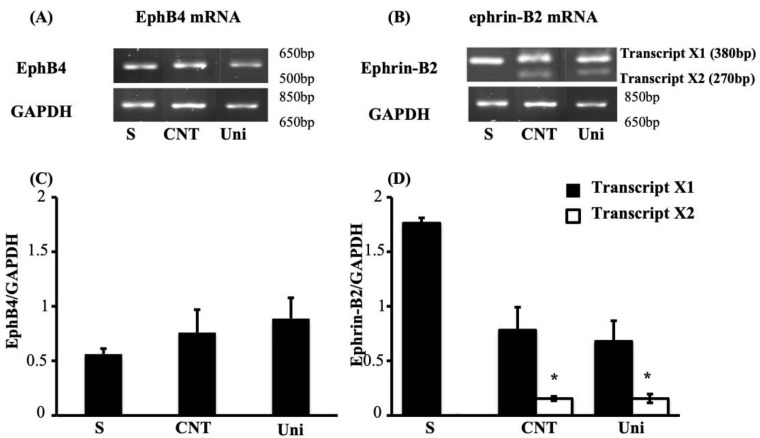
UniPR1331 did not modify colonic levels of EphB4 and ephrin-B2 mRNA. Representative agarose gels showing mRNA levels of EphB4 (**A**), ephrin-B2 (**B**) and GAPDH from the colon of vehicle-treated normal mice (S; *n* = 7) and of TNBS-treated mice administered with vehicle (CNT; *n* = 4) or UniPR1331 25 mg/kg (Uni; *n* = 5). Histograms represent the quantification of mRNA levels of EphB4 (**C**) and of transcripts X1 (black bars) and X2 (white bars), encoded by ephrin-B2 gene (**D**), normalized to GAPDH amplification products. Data are shown as mean ± SEM. * *p* < 0.05 vs. S mice, one-way ANOVA followed by Bonferroni’s post-test.

**Figure 7 pharmaceuticals-14-00502-f007:**
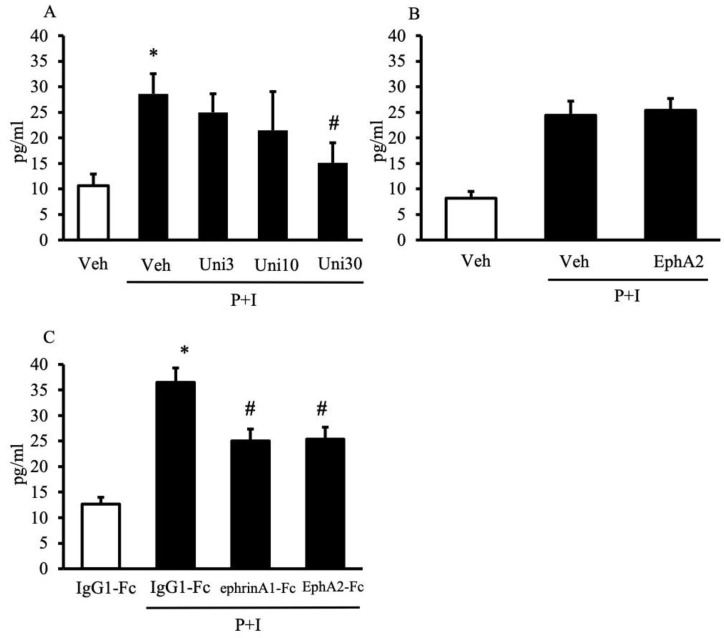
Effects of UniPR1331 and EphA/ephrin-A recombinant proteins on splenic mononuclear cells activation. TNFα levels, released by cultured splenic mononuclear cells, when incubated with: (**A**) DMSO 0.3% (Veh), UniPR1331 3–30 μg/mL (Uni3, Uni10, Uni30); (**B**) saline solution (Veh), EphA2 1 μg/mL; (**C**) IgG1-Fc 0.3 μg/mL, ephrin-A1-Fc 0.5 μg/mL, EphA2-Fc 1 μg/mL. The assays were performed in the absence (white bar) or in the presence (black bars) of PMA 50 ng/mL (P) and ionomycin 500 ng/mL (I) (*n* = 5–8 independent values per group). * *p* < 0.05 vs. unstimulated; ^#^ *p* < 0.05 vs. vehicle or IgG1-Fc in the presence of P + I; one-way ANOVA followed by Bonferroni’s post-test.

**Figure 8 pharmaceuticals-14-00502-f008:**
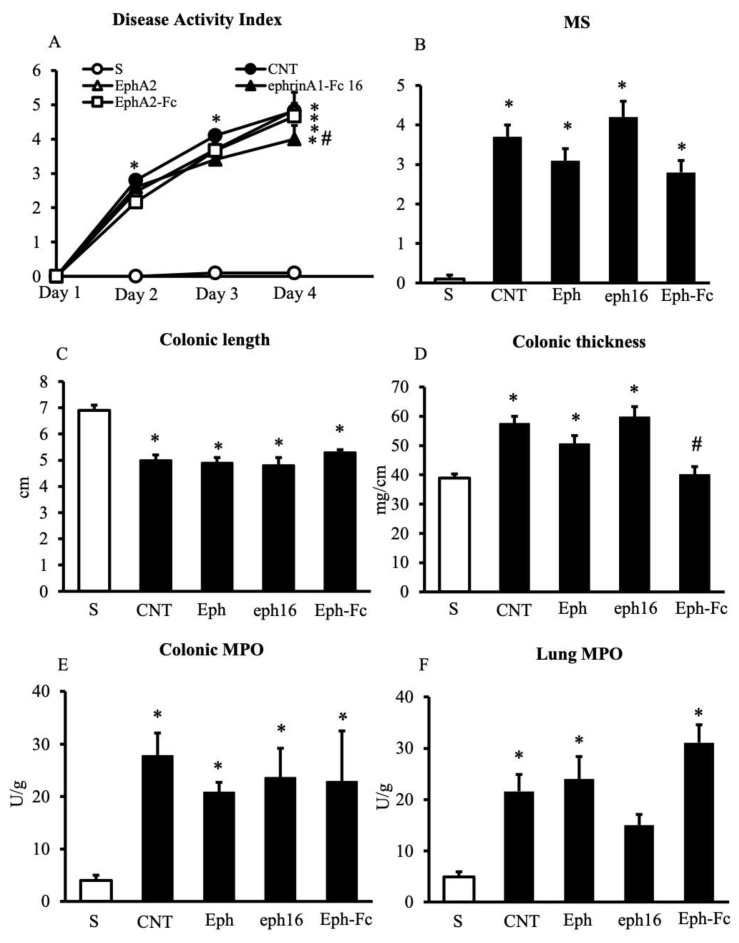
Effects of EphA/ephrin-A ligands on TNBS-induced inflammatory responses. Disease Activity Index (**A**), macroscopic score (**B**), colonic length (**C**), colonic thickness (**D**), colonic MPO (**E**) and lung MPO (**F**) activity assessed in vehicle-treated normal mice (S) and in TNBS-treated mice administered with vehicle (CNT), EphA2 20 mg/kg (Eph), ephrin-A1-Fc 16 mg/kg (eph16), and EphA2-Fc 30 mg/kg (Eph-Fc) (*n* = 5–15 independent values per group). * *p* < 0.05 vs. S mice; ^#^ *p* < 0.05 vs. CNT mice; one-way or two-way (DAI) ANOVA followed by Bonferroni’s post-test; Kruskal-Wallis followed by Dunn’s post-test (MS).

**Table 1 pharmaceuticals-14-00502-t001:** Effects of UniPR1331 and recombinant proteins on the viability of splenic lymphocytes. Percent viability of cultured lymphocytes, assessed through propidium iodide exclusion assay, after incubation with DMSO 0.3% (vehicle for UniPR1331), saline solution (vehicle for recombinant proteins), UniPR1331 (3–30 μM), IgG1-Fc 0.3 μg/mL, ephrin-A1-Fc 0.5μg/mL, EphA2-Fc 1 μg/mL and EphA2 1 μg/mL in the presence of PMA and ionomycin. Data are expressed as the mean value ± SEM (*n* = 3–4 independent values).

Treatment	Viability (%)
DMSO 0.3%	94 ± 1
Saline	95 ± 1
UniPR1331 3 μM	95 ± 1
UniPR1331 10 μM	94 ± 1
UniPR1331 30 μM	94 ± 1
IgG1-Fc 0.3 μg/mL	93 ± 1
Ephrin-A1-Fc 0.5 μg/mL	93 ± 1
EphA2-Fc 1 μg/mL	93 ± 1
EphA2 1 μg/mL	96 ± 1

**Table 2 pharmaceuticals-14-00502-t002:** Effects of EphA/ephrin-A ligands on TNBS-induced changes of spleen and MLN T cells count.

	SPLEEN			MLN		
Treatment	%CD4+	%CD8+	CD4/CD8	%CD4+	%CD8+	CD4/CD8
S	55.5 ± 2.1	36.6 ± 1.8	1.6 ± 0.1	59.3 ± 2.0	36.2 ± 1.8	1.7 ± 0.1
CNT	38.8 ± 1.2 *	50.3 ± 1.3 *	0.8 ± 0.0 *	45.2 ± 1.8 *	48.4 ± 1.7 *	1.0 ± 0.1 *
Eph	40.0 ± 1.8 *	49.4 ± 2.2 *	0.8 ± 0.0 *	46.1 ± 1.6 *	44.6 ± 3.5	1.1 ± 0.1 *
eph16	41.4 ± 2.1 *	48.4 ± 2.4 *	0.9 ± 0.1 *	45.2 ± 1.8 *	48.4 ± 1.7 *	1.0 ± 0.1 *
eph50	36.7 ± 1.8 *	53.0 ± 5.0 *	0.8 ± 0.0 *	44.0 ± 3.2 *	49.7 ± 3.7 *	1.0 ± 0.2 *
Eph-Fc	46.6 ± 3.2 *	45.4 ± 4.0 *	1.1 ± 0.1 *	45.5 ± 3.5 *	51.0 ± 3.6 *	0.9 ± 0.1 *

Percentages of CD4^+^, CD8^+^, and the ratio between CD4^+^ and CD8^+^ T lymphocytes were assessed in the spleen and MLN excised from vehicle-treated normal mice (S) and TNBS-treated mice administered with vehicle (CNT), EphA2 20 μg/kg (Eph), ephrin-A1-Fc 16 and 50 μg/kg (eph16, eph50), and EphA2-Fc 30 μg/kg (Eph-Fc). * *p* < 0.05 vs. S mice, one-way ANOVA followed by Bonferroni’s post-test.

## Data Availability

Data are available within the article and associated Appendix A.

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
