# Peer review of "UniPR1331: Small Eph/Ephrin Antagonist Beneficial in Intestinal Inflammation by Interfering with Type-B Signaling"

_pharmaceuticals, 2021, doi:10.3390/ph14060502_

Round 1
Reviewer 1 Report
Overall, the paper is clearly written with the conclusion supported by the presented results in terms of the goal of the study.
The limitation is that the study could not specifically address the role of Eph/ephrin type B signaling in lymphocyte compartments in this Th1/17-dependent colitis model. The authors did not see anti-inflammatory effects in the colon (also they did not examine the T cell changes such as by immunostaining of CD3+ or CD4+ cell infiltration) but see some systemic changes in the spleen. Therefore, the effect of Uni is likely through its systemic anti-inflammatory functions. To clarify, measuring panels of inflammatory (IL-1b, TNF, IL-12, IL-6, IL-17) vs anti-inflammatory (IL-10) cytokines in the colon and serum/plasma, and the balance of T cell subsets (Th1, Th17, Treg, and Tr1) may help better understand how Uni shows the anti-inflammatory effects and elevates the significance of work by defining the role of Eph/ephrin functions.
Minor comments:
- l73. Before starting, briefly describing the experiment procedures would be helpful. Also, S, CNT, and DAI, should be spelled out or explain what they are even though they are described in the Figure legend.
- Fig. 1. What are MS and MOP?
- l142. What is the significance of the splicing variants? The data were shown but not discussed what that means.
Author Response
Responses to referee 1
The limitation is that the study could not specifically address the role of Eph/ephrin type B signaling in lymphocyte compartments in this Th1/17-dependent colitis model. The authors did not see anti-inflammatory effects in the colon (also they did not examine the T cell changes such as by immunostaining of CD3+ or CD4+ cell infiltration) but see some systemic changes in the spleen. Therefore, the effect of Uni is likely through its systemic anti-inflammatory functions. To clarify, measuring panels of inflammatory (IL-1b, TNF, IL-12, IL-6, IL-17) vs anti-inflammatory (IL-10) cytokines in the colon and serum/plasma, and the balance of T cell subsets (Th1, Th17, Treg, and Tr1) may help better understand how Uni shows the anti-inflammatory effects and elevates the significance of work by defining the role of Eph/ephrin functions.
We thank the reviewer for the thoughtful comments but we would like to point out that the first aim of the manuscript is not to investigate the role of Eph/ephrin type B signaling in lymphocytes compartments; instead, we aimed at studying primarily the effects of UniPR1331 in TNBS-induced colitis, comparing them with those produced by EphA2 and EphB4, in order to assess whether they could be ascribed to the interference with Eph/ephrin type A or type B signalling.
We believe that our results did demonstrate that UniPR1331 exerted anti-inflammatory effects also in the colon, since it reduced colonic macroscopic damage and myeloperoxidase activity, marker of neutrophils infiltration, reverted colonic shortening and mitigated IL-1b levels. We agree with the referee that the determination of the wide panel of cytokines proposed and the assessment of the types of lymphocytes recruited in the lamina propria would surely elevate the significance of the work and we will take advantage of this suggestion for a future manuscript; however, we trust that our results are robust enough to demonstrate that the anti-inflammatory actions of UniPR1331 are mediated, at least in part, by the blockade of EphB/ephrin-B pathway and not of A-type signalling.
1.l73. Before starting, briefly describing the experiment procedures would be helpful. Also, S, CNT, and DAI, should be spelled out or explain what they are even though they are described in the Figure legend.
- Fig. 1. What are MS and MOP?
We agree with the referee that a previous description of the experimental procedures would help the reader but the complete explanation of the methods applied can be found in section 4. Materials and Methods. As suggested, the abbreviations S, CNT, DAI, MS, MPO and PMA have been explained at their first appearance in the manuscript.
3.l142. What is the significance of the splicing variants? The data were shown but not discussed what that means.
According to the referee’s suggestion, we introduced an additional comment on ephrin-B2 splicing variants in lines 275-282.

Reviewer 2 Report
Carmine Giorgio et al. report innovative results concerning a lithocholic acid-based organic compound with antagonistic effects on the Eph/ephrin signaling system. The authors demonstrate that UniPR1331 has an overt anti-inflammatory effect in TNBS-induced colitis on the bowel tissue and splenic macrophages and diminishes the infiltration of neutrophils in the lung. UniPR1331 also down-regulates the splenic CD8+ cytotoxic T cells and improves the CD4/CD8 ratio. According to their results, UniPR1331 did not influence the expression of EphA2 and ephrinA1 in the colon, nor of the EphB4 and ephrinB2. This is good quality research, based on appropriate design and a versatile methodological approach, worthwhile to be published. The treatment groups had a sufficient number of experimental animals, and the group assured the objectivity of results by blinding the investigators for the research protocols. My only uncertainty refers to the treatment doses used. They mention in Materials and methods that UniPR1331 has been applied in 10 mg/kg and 25 mg/kg doses, while ephrinA1-Fc in much smaller concentrations, 16 μg/kg and EphA2 in 50 μg/kg. Which were the considerate to choose these doses? Did they calculate any molar ratio of ligands to receptors? They also mention in the Discussion that the Eph/ephrin antagonist has favorable pharmacokinetics and cite reference number [8]; what do they mean by this statement? Some brief explanations would further improve the quality of their undoubtedly manuscript. A quick language re-editing in some sentences would also be beneficial.Author Response
Responses to referee 2
Carmine Giorgio et al. report innovative results concerning a lithocholic acid-based organic compound with antagonistic effects on the Eph/ephrin signaling system. The authors demonstrate that UniPR1331 has an overt anti-inflammatory effect in TNBS-induced colitis on the bowel tissue and splenic macrophages and diminishes the infiltration of neutrophils in the lung. UniPR1331 also down-regulates the splenic CD8+ cytotoxic T cells and improves the CD4/CD8 ratio. According to their results, UniPR1331 did not influence the expression of EphA2 and ephrinA1 in the colon, nor of the EphB4 and ephrinB2. This is good quality research, based on appropriate design and a versatile methodological approach, worthwhile to be published. The treatment groups had a sufficient number of experimental animals, and the group assured the objectivity of results by blinding the investigators for the research protocols. My only uncertainty refers to the treatment doses used. They mention in Materials and methods that UniPR1331 has been applied in 10 mg/kg and 25 mg/kg doses, while ephrinA1-Fc in much smaller concentrations, 16 μg/kg and EphA2 in 50 μg/kg. Which were the considerate to choose these doses? Did they calculate any molar ratio of ligands to receptors? They also mention in the Discussion that the Eph/ephrin antagonist has favorable pharmacokinetics and cite reference number [8]; what do they mean by this statement? Some brief explanations would further improve the quality of their undoubtedly manuscript. A quick language re-editing in some sentences would also be beneficial.
We wish to thank the referee for the very positive comments of our manuscript. To clear any doubt, we would like to specify that:
-the dosage of UniPR1331 was chosen on the basis of the doses employed by Festuccia et al., 2018, while the applied doses of EphA/ephrin-A recombinant proteins were equimolar to EphB/ephrin-B proteins administered in the previous investigation by Grandi, Zini et al., 2019. These details are reported in lines 367-370;
-as regards the citation of reference 8, clarifications have been introduced in the Introduction (lines 51-52) and in the Discussion (lines 263-265).

Round 2
Reviewer 1 Report
The revised form is acceptable.
Author Response
We warmly thank the reviewer, who judged our manuscript acceptable for publication.
Reviewer 2 Report
The authors should shortly explain the UniPR1331, ephrinA1-Fc and EphA2 doses used in treatment.
Author Response
The authors should shortly explain the UniPR1331, ephrinA1-Fc and EphA2 doses used in treatment.
We are sorry for having mistaken, in the previous answer, the number of lines in the manuscript where the explanation of the doses of UniPR1331, ephrin-A1-Fc and EphA2 used was present. The explanation in the revised version of the manuscript can be found in lines 392-394. Since this comment was already present in the original version of the paper, these lines have not been tracked in the newer version.